# GWAS Enhances Genomic Prediction Accuracy of Caviar Yield, Caviar Color and Body Weight Traits in Sturgeons Using Whole-Genome Sequencing Data

**DOI:** 10.3390/ijms25179756

**Published:** 2024-09-09

**Authors:** Hailiang Song, Tian Dong, Wei Wang, Xiaoyu Yan, Chenfan Geng, Song Bai, Hongxia Hu

**Affiliations:** 1Fisheries Science Institute, Beijing Academy of Agriculture and Forestry Sciences & Beijing Key Laboratory of Fisheries Biotechnology, Beijing 100068, China; songhailiang@baafs.net.cn (H.S.); yinghua328@163.com (T.D.); raywang8848@163.com (W.W.); yanxiaoyu21@126.com (X.Y.); gengchenfishery@163.com (C.G.); baisong@baafs.net.cn (S.B.); 2Key Laboratory of Sturgeon Genetics and Breeding, Ministry of Agriculture and Rural Affairs, Hangzhou 311799, China; 3National Innovation Center for Digital Seed Industry, Beijing 100097, China

**Keywords:** sturgeon, whole-genome sequencing, genome-wide association study, candidate gene, genomic prediction

## Abstract

Caviar yield, caviar color, and body weight are crucial economic traits in sturgeon breeding. Understanding the molecular mechanisms behind these traits is essential for their genetic improvement. In this study, we performed whole-genome sequencing on 673 Russian sturgeons, renowned for their high-quality caviar. With an average sequencing depth of 13.69×, we obtained approximately 10.41 million high-quality single nucleotide polymorphisms (SNPs). Using a genome-wide association study (GWAS) with a single-marker regression model, we identified SNPs and genes associated with these traits. Our findings revealed several candidate genes for each trait: caviar yield: *TFAP2A*, *RPS6KA3*, *CRB3*, *TUBB*, *H2AFX*, *morc3*, *BAG1*, *RANBP2*, *PLA2G1B*, and *NYAP1*; caviar color: *NFX1*, *OTULIN*, *SRFBP1*, *PLEK*, *INHBA*, and *NARS*; body weight: *ACVR1*, *HTR4*, *fmnl2*, *INSIG2*, *GPD2*, *ACVR1C*, *TANC1*, *KCNH7*, *SLC16A13*, *XKR4*, *GALR2*, *RPL39*, *ACVR2A*, *ADCY10*, and *ZEB2*. Additionally, using the genomic feature BLUP (GFBLUP) method, which combines linkage disequilibrium (LD) pruning markers with GWAS prior information, we improved genomic prediction accuracy by 2%, 1.9%, and 3.1% for caviar yield, caviar color, and body weight traits, respectively, compared to the GBLUP method. In conclusion, this study enhances our understanding of the genetic mechanisms underlying caviar yield, caviar color, and body weight traits in sturgeons, providing opportunities for genetic improvement of these traits through genomic selection.

## 1. Introduction

Sturgeons, with 27 species distributed across the Northern Hemisphere, are ancient fish that represent a remarkable evolutionary relic, often referred to as “living fossils” [1]. All of these species are listed as Appendix II species under the Convention on International Trade in Endangered Species of Wild Fauna and Flora (CITES). In the sturgeon breeding programs, there are three key traits, including caviar yield, caviar color, and body weight. Caviar yield has a direct correlation with caviar and fry production, leading to significant demand from enterprises for sturgeon breeding populations with high caviar yields. Among the range of caviar products, golden caviar is more costly than caviar of other colors, possibly due to the association of gold with luxury and quality. Additionally, sturgeon is highly valued for its meat quality, and cultivating a fast-growing sturgeon breeding population can expedite sturgeon production in the meat industry market. Therefore, it is recommended that caviar yield, caviar color, and body weight should be the primary objectives of sturgeon cultivation in China. Due to the late sexual maturity of sturgeon, which typically takes about 6–8 years, the breeding cycle is prolonged, and traditional pedigree-based methods suffer from lengthy generation intervals and low efficiency. Therefore, molecular marker-based breeding, especially genomic selection [2], emerges as an effective approach to expedite genetic progress in sturgeon breeding. However, at present, no reports exist on comprehensive genome-wide scanning for key molecular markers associated with all three traits based on population size.

Currently, with the rapid development of whole-genome sequencing technology and the reduction of costs, GWAS has gradually become a mainstream strategy for genetic analysis and identification of important candidate genes related to economic traits in livestock [3], plants [4], and aquatic animals [5]. In aquaculture, GWAS have been used for genetic dissection of meat quality in common carp [6], Atlantic salmon [7] and large yellow croaker [8], growth in catfish [9,10], disease resistance in Atlantic salmon [11] and large yellow croaker [12,13]. However, there have been no reports on GWAS for traits such as caviar yield, caviar color, and body weight in sturgeons.

The concept of genomic selection (GS) was initially proposed by Meuwissen et al. [2] in 2001. This method involves deriving genomic estimated breeding values (GEBV) from high-density markers across the entire genome, premised on the assumption that at least one SNP is in LD with quantitative trait loci (QTLs) affecting the target trait. In recent years, a growing body of research on GS in aquaculture animals has positioned it as a cutting-edge technology in aquaculture breeding, highlighting its potential to expedite breeding cycles and reduce associated costs. Enhancing the accuracy of genomic prediction is a prevalent challenge in GS, and incorporating prior information from GWAS to enhance this accuracy has been reported in rainbow trout [14], dairy cattle [15] and pigs [16]. However, many studies have found that incorporating GWAS prior information does not improve the accuracy of genomic prediction [17,18,19]. Therefore, this study implemented a unique strategy that combines LD-pruned markers and GWAS prior information to improve the accuracy of genomic prediction for caviar yield, caviar color, and body weight traits in sturgeons.

## 2. Results

### 2.1. Whole-Genome Sequencing and SNP Calling

Whole-genome sequencing was conducted for 673 fish, yielding a total of 67.84 billion reads with an average of 0.10 billion reads per individual. Among these, 92.60% of reads successfully aligned to the reference genome, resulting in an average sequencing depth of 13.69× for 673 individuals (ranging from 5.06× to 25.85×). After stringent quality control, a total of 10.41 million SNPs were identified. Figure 1 illustrates the histogram of SNP distribution and SNP density plots across all chromosomes. The number of high-quality SNPs per chromosome varied from 385 (Chr60) to 794,151 (Chr1) (Figure 1B), with an average density of 5345.31 SNPs/Mb (Figure 1A).

### 2.2. Population Structure Analysis

Through the first three principal component analyses (Figure 1C), it can be observed that individuals have similar genetic backgrounds in the Russian sturgeon population, which is beneficial for conducting GWAS. The pattern of LD, as depicted in Figure 1D, indicates that the average genome-wide LD (r2) obtained based on adjacent pairs of markers was 0.049 and the LD decay was 20 kb at r2 = 0.05, suggesting that candidate genes can be effectively mapped in GWAS results by setting the region of 20 kb upstream and downstream of significant SNPs.

### 2.3. Phenotype Statistics and Heritability Estimation

Descriptive statistical data for the analysis of traits in the Russian sturgeon population are shown in Table 1. The mean (standard deviation) caviar yield, caviar color, and body weight were 0.19 (0.057), 2.453 (0.653), and 19.933 (4.029), respectively. Coefficients of variation were high for caviar yield, caviar color, and body weight, 30.00%, 26.62%, and 20.21%, respectively. In addition, as shown in Table 2, SNP-based heritability was estimated through genome-wide association analysis, with heritabilities for caviar yield, caviar color, and weight being 0.497, 0.614, and 0.627, respectively, indicating moderate to high levels of heritability for each trait, which is advantageous for selective breeding programs.

### 2.4. Genome-Wide Association Study

Manhattan plots of GWAS for the three traits and corresponding QQ plots are shown in Figure 2. For caviar yield, no genome-wide significant SNPs were detected, and 31 SNPs reached the suggestive significant level (Figure 2A). The 31 suggestive significant SNPs were located on chromosomes 1, 2, 4, 6, 8, 9, 10, 12, 19, 22, 44, 46, and 56. For caviar color (Figure 2C), 1 genome-wide significant SNP and 36 suggestive significant SNPs were observed. The 1 genome-wide significant SNP was located on chromosome 22, and the 36 suggestive significant SNPs were located on chromosomes 1–7, 9, 12, 15, 17–22, 25, 36, 39, 45, and 51. For body weight (Figure 2E), 1 genome-wide significant SNP was detected, it was located on chromosome 12, and 225 SNPs at the suggestive significant level were observed, with 198 SNPs located on chromosome 12 and the remaining SNPs located on chromosomes 1, 2, 4–7, 10, 15–17, 21, 37, 41, 45, and 56.

The QQ plots show that the influence of population stratification was negligible (Figure 2B–D). Moreover, the average genomic inflation factors λ for the three traits were close to 1 (0.99, 0.98, and 0.99 for caviar yield, caviar color, and body weight, respectively). The QQ plots λ suggest that there were little or no residual population structure effects on the test statistic inflation.

### 2.5. Identification of Candidate Genes

GWAS based on whole-genome sequencing data were used to detect candidate functional genes. Based on the functional annotation analysis, candidate genes were detected within a 20-kb region, centering each significant and suggestive SNPs. As shown in Table 3, 29 genes were found for caviar yield, of which 10 genes were potential candidate genes. For caviar color (Table 4), 22 genes were detected, and 6 genes had functions related to caviar color. For body weight (Table 5), 77 genes were detected, of which 15 genes were potential candidate genes.

### 2.6. Genomic Prediction Performance

To assess the effect of incorporating GWAS results on genomic prediction, the accuracy of genomic prediction for caviar yield, caviar color, and body weight traits was evaluated using the GBLUP, GLDBLUP, and GFBLUP methods, as shown in Figure 3. GBLUP and GLDBLUP produced similar predictive accuracy, demonstrating that reducing SNP density to 50 K by LD pruning can yield prediction accuracy comparable to utilizing all markers. Additionally, GFBLUP produced the highest predictive accuracy in all cases, with GFBLUP improving by 2%, 1.9%, and 3.1% over GBLUP for caviar yield, caviar color, and body weight, respectively. For prediction bias, as shown in Figure 3, GFBLUP produces similar or lower prediction bias compared to GBLUP and GLDBLUP methods, e.g., for the body weight trait, the prediction biases of GFBLUP, GBLUP, and GLDBLUP are 0.269, 0.325, and 0.323, respectively. For MSE, GLDBLUP and GFBLUP produced lower values than GBLUP for caviar yield, while for the other two traits, all three methods produced similar MSE. Additionally, all three methods produced similar MAE in all cases.

## 3. Discussion

In this study, we conducted GWAS and identified several candidate genes related to caviar yield, caviar color, and body weight in Russian sturgeon. Furthermore, to verify the reliability of GWAS results, we evaluated the accuracy of genomic prediction for the three traits by combining LD pruning markers and GWAS prior information. The result showed that combining LD-pruned markers and GWAS prior information could improve the accuracy of genomic prediction for caviar yield, caviar color, and body weight traits in sturgeons.

### 3.1. Potential Candidate Genes for Caviar Yield

For caviar yield, a number of candidate genes located within 20 kb of genome-wide significant and suggestive significant SNPs were identified in both lines. Among them, the *TFAP2A* gene plays a vital role in mouse oocyte maturation [20]. Overexpression of *TFAP2A* may upregulate p300, increasing levels of histone acetylation and lactylation, which in turn impede spindle assembly and chromosome alignment, ultimately hindering nuclear meiotic division in mouse oocytes [20]. Niu et al. [21] reported that the *RPS6KA3* gene was associated with reproduction pathways in Xiang pigs. The presence of *CRB3* in many organs and its distribution pattern during mouse embryonic development suggest that the *CRB3* plays a significant role in establishing and maintaining polarity in mouse embryos [22]. For *TUBB* gene, Zhao et al. [23] reported that *TUBB* regulates spindle assembly and chromosome dynamics during mouse oocyte maturation. A study showed a role for the *H2AFX* gene in germ cell loss, and histone *H2AFX* links meiotic chromosome asynapsis to prophase I oocyte loss in mammals [24]. The *morc3* gene was related to the regulation of animal reproduction, and the deletion of *morc3* reduced the pregnancy rate of male mice and led to low fertility [25]. The *BAG1* gene was found to have potential efficacy in terms of ameliorating oocyte maturation [26]. A study showed that *RANBP2* acts as an inhibitor of premature maturation-promoting factor activation and the untimely degradation of securin in oocyte maturation, thereby preserving the accurate timing of the resumption of maturation and meiotic progression in mouse oocytes [27]. The *PLA2G1B* gene was found to be possible a newly discovered component affecting the efficacy of horse IVM/IVF [28]. A study observed that *NYAP1* plays a key role in ovarian development by regulating target genes related to the oxytocin signaling pathway, and its differential expression level in Han sheep may contribute to improving fecundity [29].

### 3.2. Potential Candidate Genes for Caviar Color

For caviar color, within the range of 20 kb of the genome wide significant and suggestive significant SNPs, only two genes, *SRFBP1* and *INHBA*, have been reported to be directly associated with pigment formation. The *SRFBP1* gene was reported to be associated with skin pigmentation in an Ogye x White Leghorn F2 chicken population [30]. The *INHBA* gene strongly controls skin pigmentation and also influences serum vitamin D levels in African Americans [31]. Surprisingly, the functions of other genes identified are directly related to immunity rather than pigment formation. Among them, the *NFX1* protein was found to encode a repressor of gene expression, suggesting that *NFX1* limits the immune response following infection [32]. Fiil et al. [33] reported that *OTULIN* restricts Met1-Ub formation after immune receptor stimulation to prevent unwarranted proinflammatory signaling. The *PLEK* gene was related to the immune system, suggesting an inactive immune regulation [34]. The *NARS* gene plays a role in oxidative stress/hypoxia and endoplasmic reticulum stress/unfolded protein response, and its mutation leads to melanoma susceptibility [35]. This suggests that the immune response, as a protective mechanism, will indirectly lead to the formation of pigment. Similar results have been reported in a large number of studies, e.g., Linher-Melville and Li [36] demonstrated that the melanocytes could swallow exogenous beads and then recruit immune cells to protect from injury in zebrafish (*Danio*
*rerio*). Similarly, the *INHBA* gene participates in the biological processes related to pigmentation [31] and also participates in the biological processes significantly related to hematopoiesis and immune system [34].

### 3.3. Potential Candidate Genes for Body Weight

For body weight, 15 potential candidate genes have been identified within the range of 20 kb of the genome-wide significant and suggestive significant SNPs. The *ACVR1* gene was identified in multiple regions and belongs to the transforming growth factor (TGF)-β superfamily, which can inhibit muscle differentiation [37]. Zhao et al. [38] reported that the *ACVR1* gene might contribute to later myogenesis and more muscle fibers in Landrace (LR, lean) than Lantang (LT, obese) pig breeds. A study indicated that a synonymous mutation g.101220 C > T located on the fifth intron of the ovis *HTR4* gene was detected, and association analysis showed that this mutation was significantly associated with growth traits in sheep [39]. The *fmnl2* gene is a candidate gene responsible for facioscapulohumeral muscular dystrophy, and it is critical for muscle development [40]. The polymorphism of the *INSIG2* gene is associated with increased subcutaneous fat in women and poor resistance training response in men [41]. The *GPD2* gene could catalyze the esterification of fatty acids to triglycerides [42]. *ACVR1C* is one of the type I transforming growth factor-β (TGF-β) receptors, and can be used as an adipocyte developmental marker [43]. The *TANC1* gene is essential for mammalian myoblast fusion [44]. Xie et al. [45] reported that *KCNH7* is the candidate gene related to growth in Licha Black Pig. A study showed that loss of the *SLC16A13* gene increases mitochondrial respiration in the liver, leading to reduced hepatic lipid accumulation and increased hepatic insulin sensitivity in high-fat diet-fed *SLC16A13* knockout mice [46]. The *XKR4* gene is related to feed intake and average daily gain of cattle [47]. SNPs near the *XKR4* gene are also associated with subcutaneous, which has been considered as a candidate for carcass traits [48]. The *GALR2* gene is a regulator of insulin resistance, and activation of *GALR2* represents a promising strategy against obesity-induced insulin resistance [49]. The *RPL39* is a crucial candidate gene associated with growth in farm animals [50]. Goh et al. [51] reported that *ACVR2A* directly and negatively regulates osteoblasts’ bone mass through activin receptor signaling. Dong et al. [52] reported that the *ADCY10* gene could be one of the key regulating switches for the energy metabolism in Yili goose. The *ZEB2* gene was also reported to be associated with body weight in Hu sheep [53].

### 3.4. Genomic Prediction Incorporating GWAS Prior Information

Whole-genome sequencing data includes most causal mutations that affect traits of interest, making genomic prediction less limited by the LD between SNPs and causal mutations. Simulation studies have shown that whole-genome sequencing data can improve the accuracy of genomic prediction within populations by 40% [54]. However, a substantial amount of empirical data suggests that whole-genome sequencing does not always provide greater prediction accuracy compared to SNP chips [17]. The primary reason is the presence of a large number of noisy loci in the genome, which adversely affect the accuracy of genomic prediction. Therefore, some studies have reported that LD pruning of whole genome sequencing data can reduce the number of noisy loci and improve the accuracy of genomic prediction [17,55,56]. However, our previous research has shown that using LD pruning to reduce SNP density to different levels cannot necessarily improve the accuracy of genomic prediction [57]. One possible reason is that while noisy loci are removed, functional loci may also be inadvertently eliminated, resulting in an inability to enhance prediction accuracy. In addition, there have been reports on using GWAS priors to improve the accuracy of genomic prediction [14,15,16], e.g., Yoshida and Yáñez [14] reported that the accuracy of genomic prediction can be improved using preselected variants from GWAS for growth under chronic thermal stress in rainbow trout. However, many studies have reported that utilizing prior information from GWAS does not improve the accuracy of genomic prediction [17,18,19]. This may be because, although functional sites are included in the genome, noisy sites have not been effectively removed, resulting in an inability to enhance prediction accuracy. Therefore, this study identified the advantages of both methods. Firstly, noisy loci were removed by performing LD pruning on whole genome sequencing data. Then, functional loci were screened using GWAS based on whole genome sequencing and combined with LD-pruned loci. The results showed that all three traits—caviar yield, caviar color, and body weight—could achieve improved accuracy in genomic prediction, further verifying the reliability of the GWAS results in this study. This study provides a new approach for enhancing the accuracy of genomic prediction based on whole-genome sequencing data.

## 4. Materials and Methods

### 4.1. Population and Phenotyping Measurement

The Russian sturgeons used in this study were from Hangzhou Qiandaohu Xunlong Sci-tech Co., Ltd. (Hangzhou, China). Details regarding fish rearing and phenotyping procedures have been provided in our previous study [58]. In 2012, 6 dams and 26 sires were artificially inseminated to create 26 full-sib families. At the age of 8, the developmental status of fish roe was assessed using in vitro puncture. Fish with an average roe diameter exceeding 2.8 mm were individually tagged with passive integrated transponder (PIT) electronic markers, and a fin sample was collected and preserved in absolute ethanol. Subsequently, these tagged fish were processed for caviar production at Hangzhou Qiandaohu Xunlong Sci-tech Co., Ltd. The body weight (BW), total caviar weight (CW), and caviar color (CC) of each fish were recorded. Caviar yield (CY) was calculated relative to the female body weight using the formula CY = CW/BW. A subjective color score for the caviar was assigned based on color depth, ranging from 1 to 4, with gold receiving a score of 4, light as 3, middle as 2, and black as 1. All caviar color scores were recorded by the same operator, who used the image as a reference guide for classification. In total, 673 fish with phenotype records were selected for subsequent analysis. The descriptive statistics of phenotypes are presented in Table 1.

### 4.2. Whole-Genome Sequencing

Genomic DNA extraction followed the standard phenol-chloroform method. Whole-genome sequencing was conducted for 673 fish on the BGI-T7 platform. Libraries were constructed, and sequencing was performed using 150 bp paired-end reads on DNBSEQ-T7 (MGI Technology Co., Ltd., Shenzhen, China). After sequencing, raw reads with a minimum average quality greater than 20 were subjected to trimming. Reads passing the filtering step were aligned against the reference genome of sterlet (*Acipenser ruthenus*) assembly ASM1064508v1 [1] using the Burrows-Wheeler Alignment (BWA, version 0.7.17) [59]. The alignment files were then converted to BAM format using SAMtools (version 1.2) [60]. To eliminate potential PCR duplicates, Picard MarkDuplicates (http://broadinstitute.github.io/picard/, accessed on 26 October 2023) was employed. SNP calling was performed using the UnifiedGenotyper utility of GATK (version 3.5) [61]. Variants were subsequently filtered using GATK Variant Filtration (version 3.5) with the following criteria: DP (Depth) ≥ 4, FS (FisherStrand) < 60, QUAL (Quality) ≥ 50, and QD (Quality by Depth) ≥ 2.0. Further details on the 673 sequenced fish are available in Appendix A.

### 4.3. Genotype Imputation and Population Structure Analysis

Imputation for missing genotypes of whole-genome sequencing data was performed with Beagle (version 4.1) [62]. Variants with a minor allele frequency (MAF) lower than 0.05 and deviation from the Hardy-Weinberg equilibrium (HWE) (*p* value < 10^−7^) were excluded using the PLINK software (version 1.90) [63]. Furthermore, due to the high level of LD in the genome, most SNPs are redundant; LD pruning was performed using PLINK [63] to remove variants in high LD (r2 > 0.9). After LD pruning, 10,409,793 SNPs were retained for the whole-genome sequencing data. Principal component analysis (PCA) was performed on the genomic relationship matrix using GCTA software (version 1.25.3) [64]. This resulted in a matrix of eigenvectors in descending order that represented principal components (PCs), where PC1 had the largest eigenvalue. The overall structuring of genetic variation was visualized in a scatterplot of the top few PCs. LD between a pair of SNPs was measured as r2, and LD decay analysis based on r2 was conducted using PopLDdecay (version 3.42) [65] to assess LD patterns.

### 4.4. Genome-Wide Association Study

A single-marker regression model was implemented to detect the association of SNP with caviar yield, caviar color, and body weight traits. The model includes a random polygenic effect to account for shared genetic effects of related individuals and to control population stratification. The statistical model is described below:
**y** = **1**μ + *b***x** + **Zg** + **e**,

in which **y** is the vector of phenotypes; **1** is a vector of ones; μ is the overall mean; *b* is the average effect of the gene substitution of a particular SNP; **x** is a vector of the SNP genotype (coded as 0, 1, or 2); **g** is a vector of random polygenic effects with a normal distribution **g** ~ *N*(0, **G**σ_a_^2^), in which σ_a_^2^ is the polygenic variance and **G** is the genomic relationship matrix and was constructed using all markers following VanRaden [66]; **Z** is an incidence matrix relating phenotypes to the corresponding random polygenic effects; and **e** is a vector of residual effects with a normal distribution *N*(0, **I**σ_e_^2^), in which σ_e_^2^ is the residual variance. The software GCTA (version 1.25.3) [64] was used to fit the model.

In order to control false positives, we used 5 × 10^−8^ as a genome-wide significance level, which was also applied in human GWAS [67]. We adopted 5 × 10^−6^ as the suggestive level [68]. The Manhattan and quantile-quantile (QQ) plots were drawn with the *CMplot* package (https://github.com/YinLiLin/R-CMplot, accessed on 5 February 2024) in R (http://www.r-project.org/, accessed on 5 February 2024).

### 4.5. Functional Genomic Analysis

Functional annotation of all coding genes of *Acipenser ruthenus* was performed using eggNOG-mapper (version 2) [69], which offers higher accuracy compared to traditional sequence similarity search methods such as BLAST search, as it avoids annotating from collateral homology. Genes located in the region between the 20 kb upstream and 20 kb downstream of the significant and suggestive SNPs were retrieved for data mining.

### 4.6. Genomic Prediction Incorporating GWAS Prior Information

In order to evaluate the genomic prediction effect of caviar yield, caviar color, and body weight traits, the genomic best linear unbiased prediction (GBLUP) based on the genomic relationship matrix and genomic feature BLUP (GFBLUP) including GWAS prior information were implemented to predict GEBV for each genotyped individual.

#### 4.6.1. GBLUP

The GBLUP [66] model was used to predict the GEBV of all genotyped individuals:y=1μ+Zg+e,where y is the vector of phenotypes, μ is the overall mean, 1 is a vector of ones, g is the vector of genomic breeding values, following a normal distribution of $N(0,\;G\sigma_{g}^{2}$), where $\sigma_{g}^{2}$ is the additive genetic variance, and G is the marker-based genomic relationship matrix [66]. Z is an incidence matrix linking g to y and e is the vector of random errors, following a normal distribution of $N(0,\;I\sigma_{e}^{2}$), where $\sigma_{e}^{2}$ is the residual variance.

For GBLUP, the G was constructed using whole-genome sequencing markers. According to our previous study [57], reducing SNP density to 50 K through LD pruning yielded similar prediction accuracy to using all markers. This method is termed GLDBLUP.

#### 4.6.2. GFBLUP

The GFBLUP [70] model, which uses prior information about genomic features, is based on a linear mixed model with two random genomic effects:y=1μ+Zf+Zr+e,where y, 1, μ, and e are the same as in the GBLUP model, f is the vector of genomic values captured by genetic markers associated with a genomic feature of interest, following a normal distribution of $N(0,\;G_{f}\sigma_{f}^{2}$); r is the vector of genomic effects captured by the remaining set of genetic markers, following a normal distribution $N(0,\;G_{r}\sigma_{r}^{2}$), and Z is an incidence matrix that links f and r to y. Matrices $G_{f}$ and $G_{r}$ were constructed similarly to G, with $G_{f}$ based on significant genetic markers determined by FDR with 0.05. $G_{r}$ utilizes 50 K SNPs obtained through LD pruning, excluding the markers used in $G_{f}$. It should be noted that GWAS analysis is based only on reference data.

To assess prediction efficiency, genomic prediction was carried out through 10-fold cross-validation (CV). The genotyped individuals were randomly split into ten folds, phenotypes from one-fold (validation population) were removed from the dataset, and the remaining folds (reference population) were used to predict the GEBV in the validation population. This 10-fold CV was replicated 20 times, resulting in 20 average accuracies of genomic prediction. The validation population was the same in each replicate of 10-fold CV for all the three methods, GBLUP, GLDBLUP, and GFBLUP. Prediction accuracy was calculated as the Pearson’s correlation between phenotypic values y and GEBV for the validation individuals, i.e., r(y, GEBV). The regression coefficient of y on GEBV was used to evaluate the bias of predictions, and the bias was expressed as the absolute value of the regression coefficient minus 1, i.e., abs(1- b(y, GEBV)). In addition, mean squared error (Mse) and mean absolute error (Mae) metrics were used to compare model performance. Mse (Mae) represented the average square (absolute) of the difference between y and GEBV centered on zero.

## 5. Conclusions

In this study, the GWAS based on whole-genome sequencing was performed for caviar yield, caviar color, and body weight in Russian sturgeon. Combining the results of GWAS and bioinformatics annotation analysis, 10 genes were identified as potential candidate genes associated with the caviar yield trait; 6 genes were considered potential candidate genes related to the caviar color trait; and 15 genes were detected as potential candidate genes related to the body weight trait. In addition, combining LD-pruned markers and GWAS prior information could improve the accuracy of genomic prediction for caviar yield, caviar color, and body weight traits in sturgeons. These findings provide valuable insights into the genetic mechanisms underlying these important traits and demonstrate the potential for their genetic improvement through advanced genomic selection methods. Future studies could further enhance this understanding by integrating advanced microscopical techniques, such as transmission electron microscopy, to provide a comprehensive morpho-functional analysis of Russian sturgeons.

## Figures and Tables

**Figure 1 ijms-25-09756-f001:**
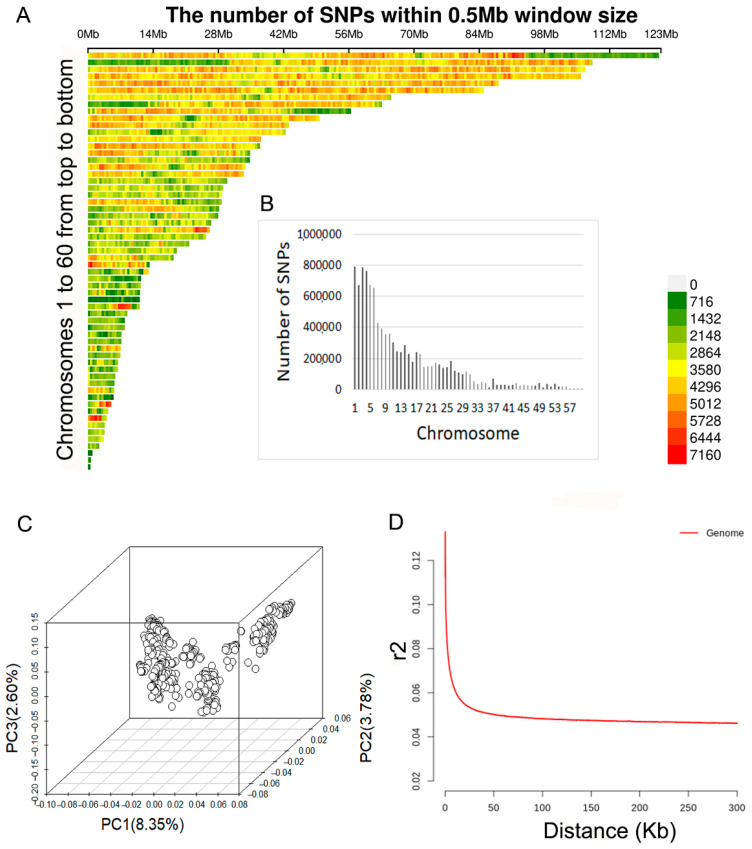
SNP distribution and population structure of Russian sturgeon. (**A**) Distribution of SNPs in 10 Mb windows across the genome; (**B**) Number of SNPs on each chromosome; (**C**) Principal component analyses for the first to the third dimensions of principal component (PC); (**D**) Genome-wide LD decay.

**Figure 2 ijms-25-09756-f002:**
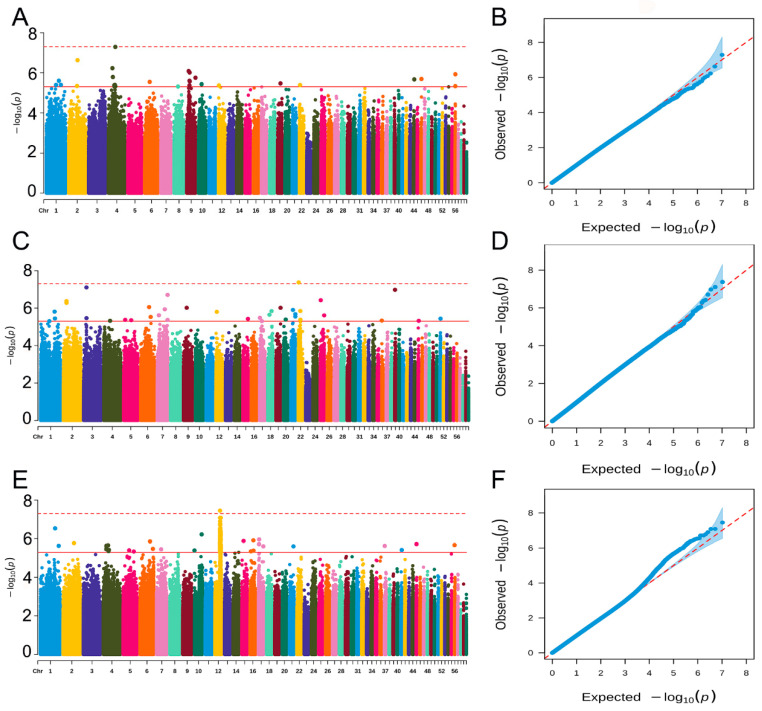
Manhattan and QQ plots of genome-wide association studies for caviar yield, caviar color, and body weight in the Russian sturgeon population. (**A**,**B**) Caviar yield; (**C**,**D**) Caviar color; (**E**,**F**) Body weight. In the Manhattan diagram, the dashed and solid lines indicate the genome-wide and suggestive significance threshold, respectively. In the Manhattan plots, different colors represent individual chromosomes. Each dot corresponds to a SNP, and its color indicates its chromosomal location.

**Figure 3 ijms-25-09756-f003:**
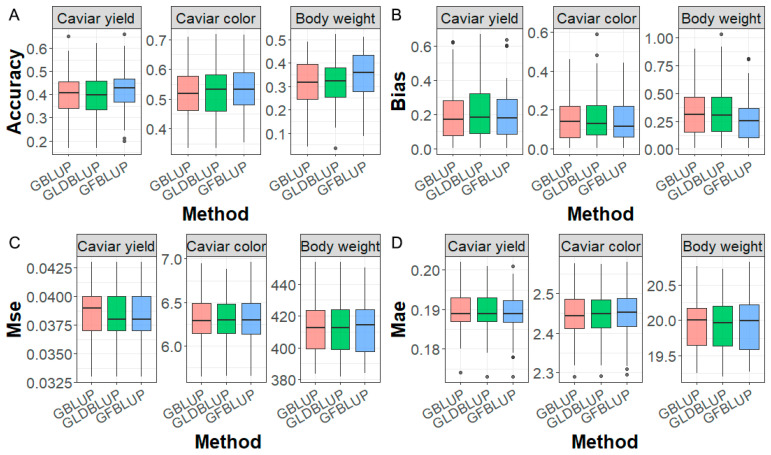
Genomic prediction performance. (**A**) Accuracy, (**B**) bias, (**C**) Mse, and (**D**) Mae of genomic prediction for caviar yield, caviar color, and body weight traits based on GBLUP, GLDBLUP, and GFBLUP methods.

**Table 1 ijms-25-09756-t001:** The descriptive statistics of caviar yield, caviar color, and body weight.

Trait	Number	Mean	SD	CV	Max	Min
Caviar yield	673	0.190	0.057	30.00%	0.439	0.021
Caviar color	673	2.453	0.653	26.62%	4	1
Body weight	673	19.933	4.029	20.21%	35.400	10.400

SD, standard deviation; CV, coefficient of variation.

**Table 2 ijms-25-09756-t002:** Estimated variance components and heritability for caviar yield, caviar color, and body weight.

Trait	V(G)	V(e)	h2
Caviar yield	0.00159	0.00161	0.497
Caviar color	0.242	0.152	0.614
Body weight	11.493	6.844	0.627

V(G), random polygenic variance; V(e), residual variance; h2, heritability.

**Table 3 ijms-25-09756-t003:** The genome significant and suggestive SNPs with the caviar yield trait using whole-genome sequencing data.

Chr	SNP_R (bp)	SNP_N	Position_Top (bp)	*p* Value_Top	Candidate Gene
4	43,712,625–43,752,625	2	43,733,654	5.12 × 10⁻^8^	* **TFAP2A** *
4	25,562,496–25,602,496	1	25,582,496	5.87 × 10⁻^7^	*C8orf34*
56	2,391,026–2,431,026	1	2,411,026	1.18 × 10⁻^6^	*PCOLCE*, *PFN2*, *RNF167*
9	55,004,692–55,044,692	1	55,024,692	1.76 × 10⁻^6^	* **RPS6KA3** *
46	4,382,490–4,422,490	1	4,402,490	2.03 × 10⁻^6^	***CRB3***, *DENND1C*, ***TUBB***
44	5,849,293–5,889,293	1	5,869,293	2.12 × 10⁻^6^	*ARCN1*, ***H2AFX***, *HMBS*
1	78,013,802–78,053,802	1	78,033,802	2.49 × 10⁻^6^	*RORB*
9	17,568,460–17,608,460	1	17,588,460	2.49 × 10⁻^6^	*SLC5A7*, *STK24*
9	19,246,701–19,286,701	1	19,266,701	2.67 × 10⁻^6^	*SETD4*, ***morc3***
4	40,684,761–40,724,761	1	40,704,761	3.98 × 10⁻^6^	***BAG1***, *C7orf25*
22	3,969,669–4,009,669	1	3,989,669	4.10 × 10⁻^6^	*DPY19L3*, *ZNF507*
4	36,848,652–36,888,652	1	36,868,652	4.20 × 10⁻^6^	*ABHD3*
9	17,393,602–17,433,602	1	17,413,602	4.25 × 10⁻^6^	* **RANBP2** *
12	2,846,164–2,886,164	1	2,866,164	4.28 × 10⁻^6^	*CRYBA4*, *CRYBB1*, ***PLA2G1B***, *TPST2*
56	1,992,379–2,032,379	1	2,012,379	4.64 × 10⁻^6^	* **NYAP1** *
8	25,489,572–25,529,572	1	25,509,572	4.83 × 10⁻^6^	*CCKBR*

Chr, chromosome. SNP_R, range of significant and suggestive SNPs region. SNP_N, number of significant and suggestive SNPs. Position_Top, the position (bp) of the top SNP in the range of significant and suggestive SNPs region. *p* value_Top, *p* value of the top SNP. The bolded text shows the potential candidate genes associated with caviar yield, identified through functional annotation with eggNOG-mapper.

**Table 4 ijms-25-09756-t004:** The genome significant and suggestive SNPs with the caviar color trait using whole-genome sequencing data.

Chr	SNP_R (bp)	SNP_N	Position_Top (bp)	*p* Value_Top	Candidate Gene
22	2,458,265–2,498,265	1	2,478,265	4.23 × 10⁻^8^	*OGFOD1*
3	16,905,242–16,945,242	1	16,925,242	7.84 × 10⁻^8^	***NFX1***, ***OTULIN***
7	63,424,745–63,464,745	2	63,444,808	1.96 × 10⁻^7^	*ALDH18A1*, *CRYGB*, *ENTPD1*
2	19,438,473–19,478,473	1	19,458,473	5.26 × 10⁻^7^	* **SRFBP1** *
6	51,347,092–51,387,092	1	51,367,092	8.82 × 10⁻^7^	*CNRIP1*, ***PLEK***
19	28,444,118–28,484,118	1	28,464,118	9.64 × 10⁻^7^	*HIC2*
21	12,773,362–12,813,362	1	12,793,362	1.26 × 10⁻^6^	*ZFYVE20*
1	85,267,483–85,307,483	1	85,287,483	1.54 × 10⁻^6^	*HCN1*
7	10,056,229–10,096,229	1	10,076,229	2.42 × 10⁻^6^	*GPR85*
25	24,937,562–24,977,562	1	24,957,562	2.45 × 10⁻^6^	*CDK16*
6	60,924,314–60,964,314	1	60,944,314	2.96 × 10⁻^6^	*FNDC4*
3	17,152,507–17,192,507	1	17,172,507	3.43 × 10⁻^6^	* **INHBA** *
1	87,476,833–87,516,833	1	87,496,833	3.57 × 10⁻^6^	* **NARS** *
51	4,080,177–4,120,177	1	4,100,177	3.66 × 10⁻^6^	*PLXNB3*
20	13,829,146–13,869,146	1	13,849,146	4.07 × 10⁻^6^	*TMEM164*
5	9,376,351–9,416,351	1	9,396,351	4.23 × 10⁻^6^	*FBXL4*
36	405,705–445,705	1	425,705	4.64 × 10⁻^6^	*APOBEC3G*
1	53,107,797–53,147,797	1	53,127,797	4.96 × 10⁻^6^	*IQCM*

Chr, chromosome. SNP_R, range of significant and suggestive SNPs region. SNP_N, number of significant and suggestive SNPs. Position_Top, the position (bp) of the top SNP in the range of significant and suggestive SNPs region. *p* value_Top, *p* value of the top SNP. The bolded text shows the potential candidate genes associated with caviar color, identified through functional annotation with eggNOG-mapper.

**Table 5 ijms-25-09756-t005:** The genome significant and suggestive SNPs with the body weight trait using whole-genome sequencing data.

Chr	SNP_R (bp)	SNP_N	Position_Top (bp)	*p* Value_Top	Candidate Gene
12	32,000,256–32,040,256	5	32,035,778	3.54 × 10⁻^8^	*BAZ2B*
12	32,955,799–32,995,799	4	32,979,846	1.95 × 10⁻^7^	*ARL6IP6*, *PRPF40A*
12	32,292,886–32,332,886	6	32,331,576	1.96 × 10⁻^7^	***ACVR1***, *UPP2*
12	33,242,523–33,282,523	5	33,281,526	1.97 × 10⁻^7^	*KALRN*
12	33,735,020–33,775,020	3	33,755,020	2.83 × 10⁻^7^	*GMPPA*, *PNKD*
12	33,222,041–33,262,041	6	33,256,286	2.90 × 10⁻^7^	*KALRN*
1	90,268,253–90,308,253	1	90,288,253	2.93 × 10⁻^7^	* **HTR4** *
12	32,383,120–32,423,120	10	32,411,784	3.09 × 10⁻^7^	*CYTIP*, *ERMN*, *GALNT5*
12	32,313,856–32,353,856	5	32,333,856	4.12 × 10⁻^7^	* **ACVR1** *
12	33,088,046–33,128,046	3	33,126,288	4.89 × 10⁻^7^	*MYLK*
12	32,359,131–32,399,131	4	32,379,835	5.00 × 10⁻^7^	*CYTIP*
12	32,989,267–33,029,267	3	33,009,267	5.21 × 10⁻^7^	* **fmnl2** *
12	33,759,093–33,799,093	3	33,779,093	5.90 × 10⁻^7^	*DARS*, *MCM6*, *PNKD*, *TMBIM1*
10	44,336,019–44,376,019	1	44,356,019	6.05 × 10⁻^7^	* **INSIG2** *
12	35,795,634–35,835,634	2	35,832,970	7.09 × 10⁻^7^	*LYPD6*
12	32,500,094–32,540,094	2	32,520,094	7.64 × 10⁻^7^	* **GPD2** *
12	32,930,958–32,970,958	6	32,959,646	7.85 × 10⁻^7^	*ARL6IP6*
12	32,849,926–32,889,926	3	32,881,001	8.75 × 10⁻^7^	*GALNT13*
12	32,870,402–32,910,402	3	32,890,482	9.58 × 10⁻^7^	*RPRM*
12	32,520,728–32,560,728	2	32,541,704	1.01 × 10⁻^6^	*NR4A2*
17	4,638,093–4,678,093	1	4,658,093	1.10 × 10⁻^6^	*ZNF536*
12	33,874,939–33,914,939	7	33,894,939	1.11 × 10⁻^6^	*THSD7B*
12	36,126,431–36,166,431	1	36,146,431	1.12 × 10⁻^6^	*UBXN4*, *enc*
12	34,918,833–34,958,833	4	34,956,534	1.20 × 10⁻^6^	*GTDC1*
16	20,101,682–20,141,682	1	20,121,682	1.23 × 10⁻^6^	*DNAJC17*
12	32,335,824–32,375,824	4	32,363,598	1.27 × 10⁻^6^	* **ACVR1C** *
15	11,074,569–11,114,569	1	11,094,569	1.32 × 10⁻^6^	*ADAP1*, *COX19*
12	31,713,257–31,753,257	2	31,747,294	1.44 × 10⁻^6^	*TBR1*
12	31,739,419–31,779,419	1	31,759,419	1.63 × 10⁻^6^	*PSMD14*, *TBR1*
12	33,184,361–33,224,361	3	33,213,805	1.79 × 10⁻^6^	*KALRN*, *ROPN1*
12	32,082,038–32,122,038	2	32,117,571	1.82 × 10⁻^6^	* **TANC1** *
12	31,495,949–31,535,949	2	31,527,074	1.83 × 10⁻^6^	* **KCNH7** *
45	2,422,442–2,462,442	1	2,442,442	1.90 × 10⁻^6^	*RNF39*, *ZKSCAN8*
56	338,814–378,814	1	358,814	2.14 × 10⁻^6^	*BCL6B*, ***SLC16A13***
12	32,031,529–32,071,529	3	32,051,529	2.14 × 10⁻^6^	*WDSUB1*
4	29,653,490–29,693,490	1	29,673,490	2.19 × 10⁻^6^	* **XKR4** *
12	36,422,230–36,462,230	1	36,442,230	2.20 × 10⁻^6^	*ESYT3*, *FAIM*
12	32,158,497–32,198,497	5	32,181,649	2.21 × 10⁻^6^	*TANC1*
4	19,975,042–20,015,042	1	19,995,042	2.29 × 10⁻^6^	*TOP1MT*
1	112,313,498–112,353,498	1	112,333,498	2.33 × 10⁻^6^	*ARID3C*
37	1,687,436–1,727,436	1	1,707,436	2.35 × 10⁻^6^	*AMIGO1*, *CYB561D1*
17	30,419,784–30,459,784	1	30,439,784	2.47 × 10⁻^6^	* **GALR2** *
21	24,163,402–24,203,402	1	24,183,402	2.48 × 10⁻^6^	*AKAP14*, *NDUFA1*, *NKAP*, ***RPL39***, *SOWAHD*, *UPF3B*
12	36,200,620–36,240,620	1	36,220,620	2.57 × 10⁻^6^	*MAP3K19*, *RAB3GAP1*
12	34,895,887–34,935,887	3	34,932,449	2.70 × 10⁻^6^	*GTDC1*
12	32,262,572–32,302,572	2	32,299,191	2.86 × 10⁻^6^	*CCDC148*, *UPP2*
12	36,763,187–36,803,187	1	36,783,187	2.90 × 10⁻^6^	*DDX18*, *Htr5b*
12	35,557,188–35,597,188	1	35,577,188	3.02 × 10⁻^6^	* **ACVR2A** *
12	34,667,954–34,707,954	1	34,687,954	3.04 × 10⁻^6^	*KYNU*
12	32,780,874–32,820,874	1	32,800,874	3.09 × 10⁻^6^	*KCNJ3*
12	32,198,180–32,238,180	2	32,218,180	3.11 × 10⁻^6^	*PKP4*, *dapl1*
12	31,634,734–31,674,734	1	31,654,734	3.19 × 10⁻^6^	***ADCY10***, *GCG*
6	79,213,869–79,253,869	1	79,233,869	3.33 × 10⁻^6^	*MARCKS*
12	31,950,349–31,990,349	1	31,970,349	3.44 × 10⁻^6^	*MARCH7*
12	36,500,960–36,540,960	1	36,520,960	3.54 × 10⁻^6^	*C2orf76*, *DBI*, *STEAP3*, *TMEM37*
12	34,988,988–35,028,988	1	35,008,988	3.69 × 10⁻^6^	* **ZEB2** *
4	34,744,439–34,784,439	1	34,764,439	3.75 × 10⁻^6^	*B4GALT6*, *TTR*
41	5,614,428–5,654,428	1	5,634,428	3.82 × 10⁻^6^	*INA*
12	33,800,017–33,840,017	1	33,820,017	4.56 × 10⁻^6^	*CXCR4*

Chr, chromosome. SNP_R, range of significant and suggestive SNPs region. SNP_N, number of significant and suggestive SNPs. Position_Top, the position (bp) of the top SNP in the range of significant and suggestive SNPs region. *p* value_Top, *p* value of the top SNP. The bolded text shows the potential candidate genes associated with body weight, identified through functional annotation with eggNOG-mapper.

## Data Availability

The datasets analyzed during this study are available from the authors upon reasonable request.

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
