# Peer review of "GWAS Enhances Genomic Prediction Accuracy of Caviar Yield, Caviar Color and Body Weight Traits in Sturgeons Using Whole-Genome Sequencing Data"

_ijms, 2024, doi:10.3390/ijms25179756_

Round 1
Reviewer 1 Report
Comments and Suggestions for Authors
The article submitted to for revision is a interesting analysis of the knowledge and understanding of how genomic is correlated with some caviar productivity like caviar colour and body weight of sturgeons. In the study, researchers performed whole-genome sequencing on 673 Russian sturgeons, renowned for their high-quality caviar. With an average sequencing depth of 13.69×, they obtained 10.41 ma high-quality single nucleotide polymorphisms (SNPs) per individual. Using a genome-wide association study (GWAS) with a single-marker regression model, they identified SNPs and genes associated with these traits. Their findings revealed several candidate genes 18 for each trait: caviar yield. Additionally, using the genomic feature BLUP (GFBLUP) method, which combines linkage disequilibrium (LD) pruning markers with GWAS prior information, they improved genomic prediction accuracy by 2%, 1.9%, and 3.1% for caviar yield and colour, and body weight traits, respectively, compared to the GBLUP method. This study enhances the understanding of the genetic mechanisms underlying caviar yield, caviar colour and body weight traits in sturgeons, providing opportunities for genetic improvement of these traits through genomic selection.
The article is well preserved and organised, and the main topic is original and well-defined. The testing hypothesis and obtained results provide an strong advancement of the current knowledge. Based on this, I think, that the manuscript fit well with the journal scope. Obtained results are interpreted appropriately and are scientifically significant. I’ve found, that all conclusions are well justified and supported by the results. In this matter, hypotheses carefully identified and tested. Used English language appropriate and understandable, I found no stylistic or technical errors. All sources are cited and the literature is well organised. Few technical small issues are marked in the attached, revisited manuscript.
The article written in an appropriate way and all presented data and analyses are presented appropriately. with the highest standards for presentation of the results used. All was presented in short, informative and precise and compact form, facilitating reception and interpretation. The whole study is correctly designed and technically sound good. Used data are robust enough to draw conclusions. All presented methods, tools, software, and reagents are described with sufficient details to allow another researcher to reproduce the results and are potentially applicable for checking. After all, I think, that the paper attract a wide readership and will be interest for a broad audience.
For sure, the work advance strongly the current knowledge. The authors address an important long-standing question with smart experiments. Those research are crucial economic traits in sturgeon breeding. Understanding the molecular mechanisms behind these traits is essential for their genetic improvement. In today's world, where obtaining high-quality caviar for the luxury market is a key issue in the protection of sturgeon. Poaching for caviar poses a threat to the last, large sturgeon, from a genetic and population point of view the most important. Studies such as the one reviewed could increase caviar productivity without affecting its quality. At the same time, it could allow more of this luxury good to be introduced to the market, which should translate into (theoretically) less poaching and illegal extraction of caviar.

Author Response
Comments 1: The article submitted to for revision is a interesting analysis of the knowledge and understanding of how genomic is correlated with some caviar productivity like caviar colour and body weight of sturgeons. In the study, researchers performed whole-genome sequencing on 673 Russian sturgeons, renowned for their high-quality caviar. With an average sequencing depth of 13.69×, they obtained 10.41 ma high-quality single nucleotide polymorphisms (SNPs) per individual. Using a genome-wide association study (GWAS) with a single-marker regression model, they identified SNPs and genes associated with these traits. Their findings revealed several candidate genes 18 for each trait: caviar yield. Additionally, using the genomic feature BLUP (GFBLUP) method, which combines linkage disequilibrium (LD) pruning markers with GWAS prior information, they improved genomic prediction accuracy by 2%, 1.9%, and 3.1% for caviar yield and colour, and body weight traits, respectively, compared to the GBLUP method. This study enhances the understanding of the genetic mechanisms underlying caviar yield, caviar colour and body weight traits in sturgeons, providing opportunities for genetic improvement of these traits through genomic selection.
Response 1: Thank you for your thoughtful review and positive feedback on our study.
Comments 2: The article is well preserved and organised, and the main topic is original and well-defined. The testing hypothesis and obtained results provide an strong advancement of the current knowledge. Based on this, I think, that the manuscript fit well with the journal scope. Obtained results are interpreted appropriately and are scientifically significant. I’ve found, that all conclusions are well justified and supported by the results. In this matter, hypotheses carefully identified and tested. Used English language appropriate and understandable, I found no stylistic or technical errors. All sources are cited and the literature is well organised. Few technical small issues are marked in the attached, revisited manuscript.
Response 2: Thank you for your thoughtful review and positive feedback on our study. Additionally, thank you for pointing out the technical issues. I have addressed them by unifying the font throughout the manuscript. Please see the newly submitted version.
Comments 3: The article written in an appropriate way and all presented data and analyses are presented appropriately. with the highest standards for presentation of the results used. All was presented in short, informative and precise and compact form, facilitating reception and interpretation. The whole study is correctly designed and technically sound good. Used data are robust enough to draw conclusions. All presented methods, tools, software, and reagents are described with sufficient details to allow another researcher to reproduce the results and are potentially applicable for checking. After all, I think, that the paper attract a wide readership and will be interest for a broad audience.
Response 3: Thank you for your thoughtful review and positive feedback on our study.
Comments 4: For sure, the work advance strongly the current knowledge. The authors address an important long-standing question with smart experiments. Those research are crucial economic traits in sturgeon breeding. Understanding the molecular mechanisms behind these traits is essential for their genetic improvement. In today's world, where obtaining high-quality caviar for the luxury market is a key issue in the protection of sturgeon. Poaching for caviar poses a threat to the last, large sturgeon, from a genetic and population point of view the most important. Studies such as the one reviewed could increase caviar productivity without affecting its quality. At the same time, it could allow more of this luxury good to be introduced to the market, which should translate into (theoretically) less poaching and illegal extraction of caviar.
Response 4: Thank you for your thoughtful review and positive feedback on our study.
Reviewer 2 Report
Comments and Suggestions for Authors
The aim of this manuscript is to obtain whole-genome sequencing data for Russian sturgeon and performed GWAS in a Russian sturgeon population to identify genomic regions and genes associated with three critical economic traits: caviar yield, caviar color, and body weight.
This manuscript is very original, shows rich content, and provides a deep insight for some works, with sufficient information. Anyway, there are some suggestions necessary to make the article complete and fully readable. For these reasons, the manuscript requires major changes.
Please find below an enumerated list of comments on my review of the manuscript:
MINOR POINTS:
There is lack of a list of the abbreviations, in this manuscript. Please, if possible, provide a list of the abbreviations, mentioned in this manuscript.
MAJOR POINTS:
INTRODUCTION:
LINE 33: Sturgeons are ancient fish, with 27 species distributed in the Northern Hemisphere. They are a remarkable evolutionary relic, earning the designation of “living fossils”. Positioned at the phylogenetic base of ray-finned fishes, sturgeons, with their archaic forms and ganoid scales, appear “frozen in time” (see, for reference: Liu, Q., & Naganuma, T. (2024). Metabolomics in sturgeon research: a mini-review. Fish Physiology and Biochemistry, 1-16).
CONCLUSIONS:
LINE 418: The development of microscopical techniques, such as transmission electron microscopy (TEM), revolutionized the morphological sciences, progressively providing new levels of magnification and resolution for exploring biological and non-biological samples. A future perspective of this study could be the project to provide a morpho-functional analysis of Russian sturgeon, with the aim to provide a molecular, structural, and ultrastructural profile of this species, by means of available and forefront microscopical techniques (see, for reference: https://doi.org/10.1016/j.aquaculture.2022.738577), as highlighted by recent scientific evidence on fishes samples. The manuscript may benefit from including this among the future perspectives of this study.
The section dedicated to methodology is very specific, by providing detailed explanation for the methods used in this study. The methodology applied is overall correct, the results are reliable and adequately discussed.
The conclusion of this manuscript is perfectly in line with the main purpose of the paper: the authors have designed and conducted the study properly. As regards the conclusions, they are well written and present an adequate balance between the description of previous findings and the results presented by the authors.
Finally, this manuscript also shows a basic structure, properly divided and looks like very informative on this topic. Furthermore, figures and tables are complete, organized in an organic manner and easy to read.
In conclusion, this manuscript provided a comprehensive analysis of current knowledge in this field. Moreover, this research has futuristic importance and could be potential for future research. However, major concerns of this manuscript are with the introductive section: for these reasons, I have major comments for this section, for improvement before acceptance for publication. The article is accurate and provides relevant information on the topic and I have some major points to make, that may help to improve the quality of the current manuscript and maximize its scientific impact. I would accept this manuscript if the comments are addressed properly.
Author Response
The aim of this manuscript is to obtain whole-genome sequencing data for Russian sturgeon and performed GWAS in a Russian sturgeon population to identify genomic regions and genes associated with three critical economic traits: caviar yield, caviar color, and body weight.
This manuscript is very original, shows rich content, and provides a deep insight for some works, with sufficient information. Anyway, there are some suggestions necessary to make the article complete and fully readable. For these reasons, the manuscript requires major changes.
Please find below an enumerated list of comments on my review of the manuscript:
MINOR POINTS:
Comments 1: There is lack of a list of the abbreviations, in this manuscript. Please, if possible, provide a list of the abbreviations, mentioned in this manuscript.
Response 1: Thank you for pointing this out. We agree with this comment. Therefore, we have added the abbreviations, please see the newly submitted version.
MAJOR POINTS:
INTRODUCTION:
Comments 2: LINE 33: Sturgeons are ancient fish, with 27 species distributed in the Northern Hemisphere. They are a remarkable evolutionary relic, earning the designation of “living fossils”. Positioned at the phylogenetic base of ray-finned fishes, sturgeons, with their archaic forms and ganoid scales, appear “frozen in time” (see, for reference: Liu, Q., & Naganuma, T. (2024). Metabolomics in sturgeon research: a mini-review. Fish Physiology and Biochemistry, 1-16).
Response 2: Thank you for pointing this out. We agree with this comment. Therefore, we have revised it, please see the introduction part.
CONCLUSIONS:
Comments 3: LINE 418: The development of microscopical techniques, such as transmission electron microscopy (TEM), revolutionized the morphological sciences, progressively providing new levels of magnification and resolution for exploring biological and non-biological samples. A future perspective of this study could be the project to provide a morpho-functional analysis of Russian sturgeon, with the aim to provide a molecular, structural, and ultrastructural profile of this species, by means of available and forefront microscopical techniques (see, for reference: https://doi.org/10.1016/j.aquaculture.2022.738577), as highlighted by recent scientific evidence on fishes samples. The manuscript may benefit from including this among the future perspectives of this study.
Response 3: Thank you for pointing this out. We agree with this comment. Therefore, we have revised the conclusions part to make it clearer, please see L412-414.
Comments 4: The section dedicated to methodology is very specific, by providing detailed explanation for the methods used in this study. The methodology applied is overall correct, the results are reliable and adequately discussed.
Response 4: Thank you for your thoughtful review and positive feedback on our study.
Comments 5: The conclusion of this manuscript is perfectly in line with the main purpose of the paper: the authors have designed and conducted the study properly. As regards the conclusions, they are well written and present an adequate balance between the description of previous findings and the results presented by the authors.
Response 5: Thank you for your thoughtful review and positive feedback on our study.
Comments 6: Finally, this manuscript also shows a basic structure, properly divided and looks like very informative on this topic. Furthermore, figures and tables are complete, organized in an organic manner and easy to read.
Response 6: Thank you for your thoughtful review and positive feedback on our study.
Comments 7: In conclusion, this manuscript provided a comprehensive analysis of current knowledge in this field. Moreover, this research has futuristic importance and could be potential for future research. However, major concerns of this manuscript are with the introductive section: for these reasons, I have major comments for this section, for improvement before acceptance for publication. The article is accurate and provides relevant information on the topic and I have some major points to make, that may help to improve the quality of the current manuscript and maximize its scientific impact. I would accept this manuscript if the comments are addressed properly.
Response 7: Thank you for your thorough review and constructive feedback. We appreciate your recognition of the importance of our research and its potential for future studies.
Reviewer 3 Report
Comments and Suggestions for Authors
This study is well conducted with clear objectives. Informative findings are well presented and discussed. I have a few questions and some minor comments which could be addressed for improving this manuscript.
Abstract: Add solid conclusion of the study.
Introduction: Please delete lines 75-79.
Results and Methods are written well.
Figure 3: Title is missing.
Tables 1-5: Table legends are missing.
Conclusions (lines 409-418): Please improve the conclusion with the solid findings of the present study.
Institutional Review Board Statement: Please add ethical approval of used fishes.
Data Availability Statement: Whole genome sequencing data did not deposit to NCBI data bank. The authors must provide accession number of submitted sequences data in the manuscript.
Author Response
This study is well conducted with clear objectives. Informative findings are well presented and discussed. I have a few questions and some minor comments which could be addressed for improving this manuscript.
Comments 1: Abstract: Add solid conclusion of the study.
Response 1: Thank you for pointing this out. We agree with this comment. Therefore, we have revised it, please see the newly submitted version.
Comments 2: Introduction: Please delete lines 75-79.
Response 2: Thank you for pointing this out. We agree with this comment. Therefore, we have deleted it, please see the newly submitted version.
Comments 3: Results and Methods are written well.
Response 3: Thank you for your thoughtful review and positive feedback on our study.
Comments 4: Figure 3: Title is missing.
Response 4: Thank you for pointing this out. We agree with this comment. Therefore, we have deleted it, please see Figure 3 and L173.
Comments 5: Tables 1-5: Table legends are missing.
Response 5: Thank you for pointing this out. The table titles and footnotes are already included within each table and the table contents have been fully addressed in the Results section of the manuscript. Therefore, to avoid redundancy, we did not include extensive legends within the tables.
Comments 6: Conclusions (lines 409-418): Please improve the conclusion with the solid findings of the present study.
Response 6: Thank you for pointing this out. We agree with this comment. Therefore, we have revised the conclusions part to make it clearer, please see L409-414.
Comments 7: Institutional Review Board Statement: Please add ethical approval of used fishes.
Response 7: Thank you for pointing this out. Our study only involves the use of fin samples for DNA extraction and sequencing, and does not involve any procedures requiring animal ethical approval. Therefore, we believe that ethical approval is "Not applicable" to this study. Please see L295 and L307.
Comments 8: Data Availability Statement: Whole genome sequencing data did not deposit to NCBI data bank. The authors must provide accession number of submitted sequences data in the manuscript.
Response 8: Thank you for pointing this out. The whole genome sequencing data have not been deposited in the NCBI data bank due to commercial application considerations. However, the datasets analyzed during this study are available from the corresponding author upon reasonable request. This approach aligns with the journal’s Instructions for Authors, which recommend the statement: "The data presented in this study are available on request from the corresponding author." Additionally, similar Data Availability Statements have been used in other articles published in this journal, such as:
https://doi.org/10.3390/ijms25116245
https://doi.org/10.3390/ijms25052626
https://doi.org/10.3390/ijms25179397
https://doi.org/10.3390/ijms25179161
Round 2
Reviewer 2 Report
Comments and Suggestions for Authors
The authors have improve the quality of the manuscript.
Reviewer 3 Report
Comments and Suggestions for Authors
The authors have made all the necessary edits. The manuscript now appears much better readable.